**New strategies for submicron characterization the carbon binding of reactive minerals in long-term contrasting fertilized soils: Implications for soil carbon storage**

**Jian Xiao[1,8], Xinhua He[2,3], Jialong Hao[4], Ying Zhou[5], Lirong Zheng[5,6], Wei Ran[1], Qirong Shen[1], Guanghui Yu[1,7]***

[1] National Engineering Research Center for Organic-based Fertilizers, Jiangsu Collaborative Innovation Center for Solid Organic Waste Resource Utilization, Jiangsu Provincial Key Lab for Organic Solid Waste Utilization, Nanjing Agricultural University, Nanjing 210095, China.

[2] Centre of Excellence for Soil Biology, College of Resources and Environment, Southwest University, Chongqing 400715, China.

[3] School of Plant Biology, University of Western Australia, Crawley, WA 6009, Australia.

[4] Key Laboratory of Earth and Planetary Physics, Institute of Geology and Geophysics, Chinese Academy of Science, Beijing 100029, China.

[5] Shanghai Institute of Measurement and Testing Technology, Shanghai 201203, China.

[6] Beijing Synchrotron Radiation Facility, Institute of High Energy Physics, Chinese Academy of Sciences, Beijing 100049, China.

[7] Department of Plant Pathology, North Carolina State University, Raleigh, NC 27695, USA.

[8] Department of Earth and Environmental Engineering, Columbia University, New York, NY 10027, USA.

* *Correspondence to*: G. H. Yu (yuguanghui@njau.edu.cn or gyu6@ncsu.edu)

**Abstract.** Mineral binding is a major mechanism for soil carbon (C) stabilization. However, the submicron information about the *in situ* mechanisms of different fertilization practices affecting organo-mineral complexes and associated C preservation remains unclear. Here, we applied nano-scale secondary ion mass spectrometry (NanoSIMS), X-ray photoelectron spectroscopy (XPS), and X-ray absorption fine structure spectroscopy (XAFS) to examine differentiating effects of inorganic versus organic fertilization on interactions between highly reactive minerals and soil C preservation. To examine such interactions, soils and their extracted colloids were collected during a 24-year long-term fertilization period (1990-2014) (no-fertilization, Control; chemical nitrogen (N), phosphorus (P) and potassium (K) fertilization, NPK; and NPK plus swine manure fertilization, NPKM). The results for different fertilization conditions showed a ranked soil organic matter concentration with NPKM > NPK > Control. Meanwhile, oxalate extracted Al ($Al_o$), Fe ($Fe_o$), short range ordered Al ($Al_{xps}$), Fe ($Fe_{xps}$), and dissolved organic carbon (DOC) ranked with NPKM > Control > NPK, but the ratios of DOC/$Al_{xps}$ and DOC/$Fe_{xps}$ ranked with NPKM > NPK > Control. Compared with the NPK treatment, the NPKM treatment enhanced the C binding loadings of Al and Fe minerals in soil colloids at the submicron scale. Furthermore, a greater concentration of highly reactive Al and Fe minerals was presented under NPKM than under NPK. Together, these submicron scale findings suggest that both the reactive mineral species and their associations with C are differentially affected by 24-year long-term inorganic and organic fertilization.

**Key words:** Al and Fe minerals; carbon binding capability; nano-scale secondary ion mass spectrometry (NanoSIMS); organo-mineral complexes; X-ray absorption fine structure spectroscopy

(XAFS); X-ray photoelectron spectroscopy (XPS)

# 1. Introduction

Associations of organic matter (OM) with pedogenic minerals, which are termed as organo-mineral complexes, are known to be key controls in the biogeochemical processes that retain OM in natural soil system (Torn et al., 1997; Kögel-knbaner et al., 2008; Mikutta et al., 2009; Schmidt et al., 2011). Soil OM (SOM) preferentially binds to rough surfaces, which provide a multitude of reactive mineral surfaces ( Chen et al., 2014; Vogel et al., 2014). These reactive minerals are also termed as short-range ordered (SRO) meta-stable colloidal minerals in volcanic ejecta (Torn et al., 1997), and serve as the nuclei for soil organic carbon (SOC) storage (Hochella et al., 2008; Kögel-Knabner et al., 2008; Remusat et al., 2012; Vogel et al., 2014). Therefore, these reactive minerals including Al and Fe minerals in soil play a critical role in determining C stability (Solomon et al., 2012; Hernes et al., 2013).

On the other hand, the reactive mineral surface of organo-mineral complexes in the complex soil matrix (mainly the top-soil layer) could be greatly improved through organic amendments to soil, such as the anthropogenic importation of organic fertilizers under long-term experimentation (Schmidt et al., 2011; Hernandez et al., 2012; Yu et al., 2012; Wen et al., 2014a; Abdala et al., 2015). Based on a meta-analysis of 49 sites and 130 observations in the world, Maillard and Angers (2014) found that cumulative manure input had a dominant effect on SOC stock changes when compared to no fertilization and chemical fertilization (Maillard and Angers, 2014). Furthermore, it was estimated that cumulative manure-C input resulted in a relative SOC change factor of $1.26 \pm 0.14$ (95% Confidence Interval (CI)). Another meta-analysis by compiling a data set of 83 studies identified the effects of improved farming practices on SOC sequestration in China (Zhao et al., 2015), and indicated that SOC

concentration and stocks at 0-30 cm depth were significantly increased under organic fertilization than under no fertilization and chemical fertilization. Although our previous results had shown that manure amendments enhanced reactive components of minerals (i.e., ferrihydrite and allophane) in soils by selective extraction methods, spectroscopies and high resolution-transmission electron microscopy (HRTEM) observation (Yu et al., 2012; Wen et al., 2014a and 2014b; Huang et al., 2016), little is known about the impacts of different fertilization practices on the *in situ* associations between reactive minerals and SOC in soil colloids at submicron scale.

In general, *in situ* investigations of natural organo-mineral complexes are restricted to bulk analyses of operationally defined physical fractions (Hatton et al., 2012 and 2015; Remusat et al., 2012; Vogel et al., 2014). In contrast, techniques for direct visualization at the submicron scale could greatly aid in gaining a better understanding of the interactions between organic structures and reactive minerals (Remusat et al., 2012; Vogel et al., 2014; Xiao et al., 2015). For instance, nano-scale secondary ion mass spectrometry (NanoSIMS) has the potential to examine the spatial integrity of soil microenvironments and has been designed for high lateral resolution (down to 50 nm) imaging, while still maintaining high mass resolution and high sensitivity (mg kg$^{-1}$ range) (Herrmann et al., 2007; Xiao et al., 2015). Previous studies have shown that NanoSIMS is an effective technique for studying natural organo-mineral complexes at the submicron scale (Herrmann, 2007; Mueller et al., 2012; Remusat et al., 2012; Hatton et al., 2012 and 2015; Vogel et al., 2014). However, NanoSIMS can not determine the morphology, elemental composition and mineral species. High-resolution transmission electron microscopy (HRTEM) combined with selected area electron diffraction (SAED) is also a promising

technique that can determine the morphology and provide detailed information on organo-mineral surfaces, as well as changes in their surface chemistries (Wen et al., 2014a; Yaron-Marcovich et al., 2005). Although direct observations of organo-mineral complexes by NanoSIMS and HRTEM have been previously described (Ramos et al., 2013; Vogel et al., 2014; Rumpel et al., 2015), few studies have reported the effects of fertilization practices on the organo-mineral complexes in soil colloids. In addition, X-ray photoelectron spectroscopy (XPS) can identify the oxidation state of elements on the surface (2~10 nm) of minerals (Zhu et al., 2014). Compared with XPS technique, X-ray absorption fine structure spectroscopy (XAFS) is a powerful tool for both identification and quantification of different mineral phases presenting in soil colloids (Li et al., 2015; Xiao et al., 2015).

Using soil colloids extracted from 24-year fertilized soils (1990-2014), the objective of this study were to address 1) the effects of fertilization practices on the quantity and composition of Al and Fe minerals, and 2) the *in-situ* interactions between SOC and minerals at the submicron scale.

## 2. Materials and methods

### 2.1. Soil samples

Samples of soil (Ferralic Cambisol, FAO soil taxonomic classification) were from a long-term (1990-2014) fertilization site in Qiyang, Hunan, Southern China (26°45′N, 111°52′E, 120 m above sea level). The long-term fertilization experiment, which belongs to the Institute of Agricultural Resources and Regional Planning, Chinese Academy of Agricultural Sciences, has been under an annual rotation of wheat and corn cropping system since September 1990. The topsoil contained 61.4% clay, 34.9% silt and 3.7% sand. Three fertilization treatments with 2 replicates or plots (20 m × 10 m) for each treatment

were examined as follows: 1) no fertilization (Control), 2) chemical nitrogen (N), phosphorus (P) and potassium (K) (NPK) and 3) a combination of the chemical fertilizers with swine manure (NPKM) (see Fig. S1 for detailed fertilization rates). The NPK and MNPK had the same total application of 300 kg N ha$^{-1}$ yr$^{-1}$. The applied N was 100% urea in the NPK, but was 30% from urea with the remaining 70% from swine manure in the MNPK. A 1.0-m-deep cement buffer zone was constructed between each plot. Each soil sample was a composite of twenty random cores (5 cm internal diameter auger) collected at 0-20 cm depth from one replicate plot. The fresh soil was mixed thoroughly, air-dried, and sieved (5 mm) for further analyses.

## 2.2. Soil colloids extraction and quantitation of highly reactive Al and Fe minerals

The soil colloids extraction was based on a previously described method (Schumacher et al., 2005). Briefly, 100 g of fresh soils was suspended in 500 mL of deionized water on a horizontal shaker (170 rpm) for 8 hr at $25 \pm 1°C$, and centrifuged at 2500 $g$ for 6 min (Fig. S1). Aliquots of the supernatant suspensions and freeze-dried soil colloid samples were then generated. Quantitation of highly reactive minerals, including Al and Fe minerals (Al$_o$, Fe$_o$), was performed using the acid ammonium-oxalate extraction method (Kramer et al., 2012). In brief, soil was extracted using 0.275 M ammonium oxalate at pH 3.25 with a soil : extractant = 1 : 100 (w/v) ratio. Ammonium oxalate was used to selectively remove short-range ordered hydrous oxides of Fe and Al such as ferrihydrite and allophane.

## 2.3. HRTEM analysis

HRTEM samples were prepared by dropping soil mobile colloids onto carbon coated copper grids. The images were recorded at an acceleration voltage of 200 keV using a JEOL JEM-2100F microscope

(JEOL JEM-2100F, Japan), which was at the Analysis and Testing Centre of Nanjing Normal University, China. HRTEM images, selected area electron diffraction (SAED) and energy dispersive X-ray analysis (EDX) were conducted using the JEOL JEM-2100F microscope to characterize soil colloid samples.

### 2.4. NanoSIMS analyses

For NanoSIMS measurements, several aliquots of the colloidal suspension from these three different fertilization treatments were separately dropped onto a silicon wafer and air-dried. In this study, we achieved 6 NanoSIMS images for each soil colloid sample to support the replicates of our results (Philippe and Schaumann, 2014; Xiao et al., 2015). For every sample, all 6 spots were analyzed to obtain a reliable data basis for the calculation of the fate of $^{12}C^-$, $^{27}Al^{16}O^-$, and $^{56}Fe^{16}O^-$ (Table S2).

The analyses were performed on a NanoSIMS 50L (Cameca, Gennevilliers, France) instrument at the Institute of Geology and Geophysics, Chinese Academy of Sciences, Beijing, China. Prior to analysis, the gold coating layer (30 nm) and any possible contamination of the sample surface were sputtered using a high primary beam current (pre-sputtering). During the pre-sputtering, the reactive $Cs^+$ ions were implanted into the sample to enhance the secondary ion yields. The primary beam (~0.9 pA) focused at a lateral resolution of 100-200 nm, was scanned over the samples, and the secondary ion images of $^{12}C^-$, $^{27}Al^{16}O^-$, and $^{56}Fe^{16}O^-$ were simultaneously collected by electron multipliers with an electronic dead time fixed at 44 ns. The presence of $^{12}C^-$ ion mass indicated SOC, while the presence of $^{27}Al^{16}O^-$ and $^{56}Fe^{16}O^-$ demonstrated Al and Fe minerals, respectively. We compensated for the charging due to the non-conductive mineral particles using the electron flood gun of the NanoSIMS instrument.

All measurements were conducted in an imaging mode. The dwell time was 1 ms pixel$^{-1}$ for all acquisitions. In this study, the NanoSIMS image sizes were 256 × 256 pixels, Control, no fertilization, 28 × 28 μm$^2$; NPK, chemical nitrogen, phosphorus and potassium fertilization, 30 × 30 μm$^2$; NPKM, chemical NPK plus swine manure fertilization, 25 × 25 μm$^2$, respectively. Specific and relevant details describing NanoSIMS measurements can be found in previous publications (Vogel et al., 2014; Rumpel et al., 2015; Xiao et al., 2015).

The analyses were carried out on different cluster compositions using the Image J software with the OpenMIMS plugin (http://www.nrims.hms.harvard.edu/ NRIMS_ImageJ.php). In this study, regions of interest (ROIs) were selected according to the intensity of the secondary $^{12}C^-$ ion mass. After pre-sputtering, since all measurements on the NanoSIMS instruments were done in an image mode, the pixels on the images from the lowest to the highest could show the intensity of the secondary $^{12}C^-$ ion mass and also represent the real distribution and heterogeneity of soil MOAs (Herrmann et al., 2007; Mueller et al., 2012; Rumpel et al., 2015; Vogel et al., 2014). In this study, the visible SOC surface areas were divided into rich and less rich $^{12}C^-$ ROIs according to the pixel value extracted from the NanoSIMS images. The $^{12}C^-$ rich ROIs included the areas above 90 pixels and the $^{12}C^-$ less rich ROIs included the areas in the range of 90-40 pixels under Control and NPK, while the $^{12}C^-$ rich ROIs were above 50 pixels and the $^{12}C^-$ less rich ROIs were in the range of 50-30 pixels under NPKM. The threshold option of the Image J software was used to automatically generate the ROIs from these NanoSIMS images. In doing so, the triangle algorithm was used (Vogel et al., 2014; Xiao et al., 2015). The same ROIs were simultaneously selected in the $^{27}Al^{16}O^-$ and $^{56}Fe^{16}O^-$ images.

## 2.5. XPS analyses

The sample preparation for the XPS procedures was adapted from Gerin et al. (2003). The XPS data were collected using a PHI 5000 Versa Probe X-ray photoelectron spectrometer (UlVAC-PHI, Japan) equipped with a monochromatized Al Kα X-ray source (1486.69 eV). The binding energy scale was corrected using the adventitious hydrocarbon C 1s spectrum (C 1s = 284.6 eV) (Zhu et al., 2014). The analyzed zone corresponded to a 300 μm × 300 μm elliptical spot. The surface charge induced by the photo ejection process was balanced using a flood gun at 6 eV. To optimize the signal to noise ratio, spectra were recorded at a detector resolution corresponding to 0.125 eV per channel. The base pressure in the spectrometer was $6.7 \times 10^{-10}$ Torr. The XPS data analyses were performed using XPSPEAK 4.1 with Shirley background correction, as referenced at http://www.lasurface.com/xps/index and http://srdata.nist.gov/xps/Default.aspx. No fixed full width at half maximum (FWHM) values were determined for the spectra of soil colloids collected under contrasting fertilization treatments. Gaussian-Lorentzian ratios were freely fit for all peaks in this study (Liang et al., 2008).

## 2.6. XAFS spectra analyses

Fe K-edge X-ray absorption fine structure (XANES) and extended X-ray absorption fine structure (EXAFS) spectra were recorded at the beamline 1W1B at the XAFS Station of the Beijing Synchrotron Radiation Facility (BSRF, Beijing, China) using a Si (111) double-crystal monochromator. The storage ring was operated at 2.5 GeV with the electron current decreasing from 240 to 160 mA within approximately 8 hrs. Samples were ground into fine powders and brushed onto tapes, which were then stacked together to yield approximately one X-ray-absorption length at their corresponding metal edges.

The intensities of incident and transmitted X-rays were monitored by ionization chambers filled with nitrogen gas. All reported spectra were measured at 20°C. Spectra were collected in quick-scan and

transmission mode. XANES spectra were recorded with 0.5 eV step, counting 10 s from 7,100 to 7,800 eV. EXAFS spectra were recorded up to k = 14.0 Å$^{-1}$, using 1 eV steps and counting for 100–200 s per scan. To improve data quality, 5 XANES scans and 5 EXAFS scans were recorded for each sample. The X-ray energy scale was calibrated to the iron K-edge (7112.0 eV) using an iron metal foil prior to XAFS acquisition. Averaged spectra were normalized using Athena (Version 2.1.1, California, USA) software,

and EXAFS data were extracted using the Autoback routine, using the same program. The spectra were normalized by subtracting a first-order polynomial fitted to the data from −100 to −30 eV before the edge and subsequently dividing through a second-order polynomial fitted to the data from 60 to 450 eV above the edge. Linear combination fitting (LCF) of XANES data were performed with the respective functions of Athena. EXAFS spectra were extracted using the Autobk algorithm (Rbkg = 0.9; k-weight

=3, spline k-range 0–11.8 Å$^{-1}$). The Fe K-edge XANES spectra with LCF of eight standard iron samples were used to precisely characterize the composition of Fe minerals (Baumgartner et al., 2013; Senn et al., 2015). The standard iron samples (either purchased or synthesized) of ferrous sulfate, ferrous oxalate, ferric sulfate, ferric oxalate, goethite, hematite, ferrihydrite, and maghemite were also recorded in a transmission mode (Table S3). A standard was considered to have a substantial contribution if it

accounted for more than 10% of a linear combination fit. The quality of the LCF was given by the residual value, the goodness-of-fit parameter $R$, defined by $R = 6[I_{exp}(E)-I_{cal}(E)]^2/6[I_{exp}(E)]^2 \times 100$ where $I_{exp}$ and $I_{cal}$ are the absorption of the experimental and calculated spectra, respectively.

**2.7. Chemical analyses**

The concentration of SOC was quantified using a CN analyzer (Vario EL, Elementar GmbH, Hanau, Germany), while SOM was $1.724 \times$ SOC. Soil pH was determined using a pH electrode at a 1:5 soil: distilled water ratio. The concentration of Fe and Al was quantified by inductively coupled plasma atomic emission spectroscopy (710/715 ICP-AES, Agilent, Australia). The concentration of DOC was determined by a TOC/TN analyzer (multi N/C 3000, Analytik Jena AG, Germany).

**2.8. Statistical analyses**

One-way analysis of variance (ANOVA) was used to test the effects of long-term fertilization on physiochemical characteristics in the soil. Significant differences between treatments (means ± SE, $n = 3$) were determined by Tukey's HSD post hoc test at $P < 0.05$, where the conditions of normality and homogeneity of variance were met.

**3. Results**

**3.1. Concentration and morphology of organo-mineral complexes in soil colloids under contrasting fertilizations**

Compared with a soil pH of 5.47 under Control, soil pH significantly ($P < 0.05$) decreased to 4.15 under NPK but significantly ($P < 0.05$) increased to 5.84 under NPKM (Table 1). SOM concentrations under different fertilizations ranked as NPKM > NPK > Control (Table 1). A general higher of the oxalate extracted Al ($Al_o$), Fe ($Fe_o$), SRO Al ($Al_{xps}$), Fe ($Fe_{xps}$), and DOC ranked as NPKM > Control > NPK, but the DOC/$Al_{xps}$ and DOC/$Fe_{xps}$ ratios ranked as NPKM > NPK > Control (Table 1).

To get insight into the spatial distribution of SOM associated with reactive mineral particles, we

used both HRTEM and NanoSIMS to acquire *in situ* observations of such associations. At the nanometer scale, the HRTEM images of extracted soil colloids provided direct visualization of the presence of soil SRO minerals from Control-, NPK- and NPKM-fertilized soil samples (Fig. 1 and Fig. S2). Soil minerals showed amorphous and crystalline patterning in different regions (Fig. 1-a, b and Figs. S2-a, b, e, f). The SAED (selected area electron diffraction) pattern further demonstrated that the amorphous mineral species were dominated by Al, Si, and O, while the crystalline minerals were mainly composed of Fe and O (Fig. 1-c, d and Figs. S2-c, d, g, h).

The NanoSIMS images of $^{12}C^-$, $^{27}Al^{16}O^-$, and $^{56}Fe^{16}O^-$ ion masses showed the submicron elemental distribution and spatial heterogeneity in the soil colloids (Figs. 2 and Fig. S3). The color bar on the NanoSIMS images, from blue to white directly showed the ion masses intensity from the relatively weak to strong at a spatial and submicron scale. Under different fertilizations, the characterization of organo-mineral complexes were randomly distributed and highly heterogeneous. Meanwhile, the arrangement and intensity were obviously various among the $^{12}C^-$, $^{27}Al^{16}O^-$, and $^{56}Fe^{16}O^-$ ion masses.

### 3.2. Binding capability of C by Al and Fe minerals

We next used the region of interests (ROIs) to explore the C binding capability of Al and Fe minerals. Fig. S3 presents a representative NanoSIMS image, which is able to show the position of region of interests (ROIs) among the several replicates (spots) of different fertilization treatments (Control: 8 replicates; NPK: 6 replicates; NPKM: 6 replicates, respectively). Based on the pixel value of secondary $^{12}C^-$ ion mass in all spots from different fertilization treatments, the selected ROIs were identified. The selected ROIs were further divided into $^{12}C^-$ rich- and $^{12}C^-$ less rich- ROIs (Fig. S3).

Table S2 lists the quantification of $^{12}C^-$ rich ($^{12}C^-$-R) and $^{12}C^-$ less-rich ($^{12}C^-$-LR) ROIs. The area percentage of the $^{12}C^-$ rich- or $^{12}C^-$ less rich- ROIs accounted for 7.47% or 40.18 %, 10.80% or 27.64 % and 8.23% or 37.99% under Control, NPK and NPKM, respectively (Table S2). The area of percentage between the Control and the NPKM was similar but different from the NPK, suggesting that compared to no fertilization control, chemical fertilization could change organo-mineral associations at the submicron scale in soil colloids, but chemical plus organic fertilization (NPKM) could eliminate the effect of chemical fertilization on organo-mineral associations. Interestingly, the box plots (Fig. 3) of $^{12}C^-/^{27}Al^{16}O^-$ (a, b) and $^{12}C^-/^{56}Fe^{16}O^-$ (c, d) ratios showed that both the median and the mean value were higher under NPKM than those under NPK. These results provided *in-situ* observation evidence at the submicron scale demonstrating that more organic C had been bound by Al and Fe minerals under NPKM than under NPK (Figs. 2 and 3), which is consistent with previous results from bulk analyses (Maillard and Angers, 2014).

**3.3. Chemical speciation of reactive minerals and C**

The XPS Al 2p$_{3/2}$ peak-fitting results (Table 2 and Fig. 4) showed that 45% allophane (~73.80 eV), 29.4% of boehmite (~74.5 eV) and 26% Al Ox (~75.40 eV) were present in soil colloids under NPKM. In contrast, approximately 43% and 34% of allophane were observed in soil colloids under NPK and Control, respectively. Considering higher (over 5 times) total Al concentrations in soil colloids under NPKM than under NPK (Table 1), the amount of allophane in soil colloids would approximately be 5 times higher under NPKM than under NPK.

Linear combination fitting (LCF) of soil colloids (Fig. S4 and Table 3) showed that goethite (56.8%-67.0%) and hematite (14.9%-25.0%) were prominent under all three fertilization treatments. The remaining Fe phases were composed of the less crystalline ferrihydrite species. The percentage of

ferrihydrite was the highest under NPKM (18.0 ± 0.02%), followed by under Control (16.0 ± 0.03%) and under NPK (6.30 ± 0.02%). In view of the better C binding and potential preservation capability of ferrihydrite when compared to goethite and hematite (Baker et al., 2010; Kramer et al., 2012; Lalonde et al., 2012; Xiao et al., 2015), it was reasonable to conclude that there was a greater C loading by Fe minerals under organic fertilization than under chemical fertilization. These results are consistent with

the previous meta-analysis (Maillard and Angers, 2014) while evidencing an *in-situ* observation at the submicron scale.

Furthermore, Fe K-edge EXAFS was used for qualitative analysis of the composition of Fe minerals in soil colloids. The Fe $k^3$-weighted EXAFS spectra (Fig. 5, left) showed that the spectral features of soils colloids under Control and NPKM were more similar to those of goethite, hematite, and

ferrihydrite than to other minerals or compounds, suggesting that those Fe minerals could be mainly composed of goethite, hematite, and ferrihydrite. The spectral features of the soils under NPK were more similar to those of goethite and hematite than to those of other minerals or compounds, supporting that those Fe minerals could be mainly composed of goethite and hematite rather than the short-range ordered ferrihydrite. Specifically, the EXAFS of Fe oxides showed double antinodes at 9.2 and 11.6 Å$^{-1}$

under Control and NPKM, whereas triple antinodes were observed under NPK at 9.2, 10.3 and 11.6 Å$^{-1}$ (Fig. 5, left). Double antinodes were found in hematite and ferrihydrite, whereas triple antinodes were

observed in goethite. These results implied that the coordination environment for Fe-Fe linkages in Control and NPKM samples might be different from that in NPK samples because the observed peak primarily derived from the Fe-Fe coordination in goethite (Mitsunobu et al., 2012).

Fourier transforms showed that Fe minerals under Control and NPKM had most of the features observed in goethite, hematite, and ferrihydrite [i.e., first peak (Fe-O) and second peak (edge-sharing Fe-Fe)] and amplitude of multiple-scattering peak at 5.2 Å. Specifically, the first shell at 1.5 Å corresponds to the Fe-O coordination, and the intensity and position were approximately identical between the Control or NPKM treated soil colloids and ferrihydrite spectra. In contrast, the second shell

identified at R + $\Delta$R = 2.3-3.5 Å corresponding to the Fe-Fe coordination was smaller than that of ferrihydrite. These results indicated that Fe in the Control and NPKM treated soil colloids might have a weaker Fe-Fe linkage than that in ferrihydrite.

In addition, the XPS C 1s peak-fitting results (Table 2 and Fig. 4) demonstrated that aromatic C (Ar-C-C/Ar-C-H, ~284.6 eV) was dominant under all three fertilizations, with the highest percentage

(75.86%) under NPKM, followed by NPK (62.51%) and Control (62.26%). In contrast, the percentages of other carbon groups, i.e., ether or alcohol carbon (C-O) and ketone or aldehyde carbon (C=O), were lowest under NPKM among the three contrasting fertilization treatments.

**4. Discussion**

**4.1. Long-term organic fertilization increased the concentration of highly reactive Al and Fe**

**minerals and their soil C binding capacity**

The selective extraction method (Table 1) showed that organic fertilization increased 36.36% of highly reactive Al ($Al_o$) and 33.33% of highly reactive Fe minerals ($Fe_o$) compared with the Control treatment, but increased 63.64% of $Al_o$ and 46.67% of $Fe_o$ compared with the NPK treatment. Therefore, organic fertilization facilitates the formation of highly reactive Al and Fe minerals. This is consistent with several previous investigations about chemical extraction methods. For example, Zhang et al. (2013) observed that the oxalate-extractable Fe ($Fe_o$) content of NPKM and M treatments was greater than that of N and NPK treatments in the 20–40 cm layer, but there was no statistical differences between the manure treatments (NPKM and M) and mineral fertilizer treatments (N and NPK) at 0–20 cm. Meanwhile, the pyrophosphate-extractable Fe ($Fe_p$) concentrations were less in the NPKM and M treatments than those in the N and NPK treatments at 0–20 cm. Using the citrate-bicarbonate-dithionite (CBD) extraction method, Huang et al. (2016) showed that organic fertilization treatments (NPKM and M) increased the iron freeness index (i.e., the $Fe_d/Fe_t$ ratio) when compared to chemical fertilization treatment (NPK). In addition, Wen et al. (2014) found that compared with chemical fertilization (N and NPK), organic fertilization (NPKM and M) significantly ($P < 0.05$) increased amorphous Al and decreased exchangeable Al, while the addition of lime (N with lime and NPK with lime) significantly ($P < 0.05$) increased weakly organically bound Al and decreased exchangeable Al. By [27]Al nuclear magnetic resonance (NMR) and Fourier- transform infrared spectroscopy (FTIR) spectroscopy, Wen et al. (2014b) confirmed the presence of amorphous Al as allophane and imogolite in soil colloids under no fertilization and organic fertilization but not under chemical fertilization. However, the direct

potential of C preservation capacity by Al and Fe minerals under different fertilizations regimes remains unexplored.

In this study, the ROI analyses of NanoSIMS observation (Fig. 3) indicated that despite of highly spatial heterogeneity of organo-mineral complexes at the submicron scale, long-term organic fertilization strengthened the SOC binding and potential preservation capability of Fe minerals for both

the $^{12}C^-$ rich- or $^{12}C^-$ less rich- ROIs in soil colloids compared to chemical fertilization. Meanwhile, long-term organic fertilization also strengthened the SOC binding and potential preservation capability of Al minerals for the $^{12}C^-$ rich- ROIs in soil colloids when compared to chemical fertilization. However, as for the $^{12}C^-$ less rich- ROIs in soil colloids, fertilization regimes seemed to have no influence in the SOC binding with Al minerals. Additionally, colloids from the NPKM treated soil had higher ratios of

$DOC/Al_{xps}$ and $DOC/Fe_{xps}$ than those under Control and NPK (Table 1), which was compatible with the assumption suggested by the NanoSIMS and HRTEM. These results could be derived from a long-term continuous organic C input that might have enriched soil microbial communities and then in turn supported an efficient formation of the concomitant organo-mineral aggregates (Wild et al., 2014; Basler et al., 2015).

Moreover, it was notable that higher proportion of aromatic C (Ar-C-C/Ar-C-H) while lower proportion of ether or alcohol carbon (C-O) or ketonic or aldehyde carbon (C=O) were observed under NPKM than under NPK or Control, which indicated that additional aromatic functional groups might have a priority attaching to the highly reactive Al and Fe minerals compared with other carbon groups. This result was also supported by C 1s near-edge X-ray fine structure (NEXAFS) spectroscopy that

compared to the NPK treatment, the NPKM treatment markedly increased the percentages of both the aromatic (283.0-286.1 eV) and phenolic (286.2-287.5 eV) groups over 2.8-fold (Huang et al., 2016). Moreover, the XPS C 1s peak-fitting results (Table 2 and Fig. 4) demonstrated that the highest percentage of aromatic C (75.86%) was present under NPKM, followed by under NPK (62.51%) and under Control (62.26%). The previous investigation had shown that aromatic C in composted dairy

manure accounted for approximately 30% of the total C, taking advantage of the solid-state $^{13}$C nuclear magnetic resonance (NMR) spectroscopy (Liang et al., 1996). And the addition of manure-based amendments increased SOC and enhanced aggregate stability (Mikha et al., 2015). But it is unclear whether manure is direct contributed to aromatic C increase or first utilized by microbes and then contributed to aromatic C increase in this study. Aromatic compounds are preferentially retained at the

interface of reactive minerals and that long-term C storage by SRO minerals has occurred via the mechanism of chemical retention with dissolved aromatic acids (Kramer et al., 2012; Huang et al., 2016). These results were due to that the long-term continuous organic C input could improve the spatial arrangement within the mineral matrix (i.e., more amorphous minerals), the fine-scale redox environment (i.e., appropriate pH), microbial ecology (i.e., appropriate pH, manure) and interaction

with mineral surfaces under fertile and weakly acidic conditions (Wild et al., 2014; Basler et al., 2015; Lehmann and Kleber, 2015).

**4.2. Long-term organic fertilization modified the composition of highly reactive Al and Fe minerals**

Our results from both XPS, NanoSIMS and Fe K-edge XAFS showed that organic fertilization

facilitated the formation of highly reactive Al and Fe minerals, e.g., allophane, imogolite, and

ferrihydrite (Tables 2-3 and Figs. 3-5), which could further explain why long-term organic manure

fertilization was able to improve the C and N binding capacity of Al and Fe minerals. The data from the

TOC and ICP-AES (Table 1) also supported that soils under NPKM contained significantly higher

percentages of $Al_o$, $Fe_o$, SRO minerals, and SOM than those under NPK. The results from HRTEM and

SAED (Fig. 1) further showed that soil colloids under NPKM were composed of large amounts of

meta-stable amorphous or SRO minerals (e.g., allophane, imogolite and ferrihydrite), which could form

stable organic-mineral bonds through anion and inner-sphere ligand-exchange reactions and would thus

be well-suited to physically protecting geometries (Torn et al., 1997; Yu et al., 2012; Basler et al., 2015).

It would be an innovative method using the ratio of $^{12}C^-/^{27}Al^{16}O^-$ and $^{12}C^-/^{56}Fe^{16}O^-$ on NanoSIMS

images to quantify the stronger binding ability under NPKM compared with other fertilization

treatments (Fig. 3). These results are consistent with previous studies using the $^{27}Al$ NMR spectroscopy

and FTIR that long-term organic treatment released greater amounts of minerals into soil solutions than

chemical fertilizers (NPK) treatment (Yu et al., 2012; Wen et al., 2014a,b; Wu et al., 2014). Using the Fe

K-edge XANES spectroscopy, Huang et al. (2016) also showed that reactive Fe minerals were mainly

composed of less crystalline ferrihydrite in the organic manure-treated soil and more crystalline goethite

in the NPK-treated soil. By measuring the composition of manure, Wen et al. (2014b) had shown that

the reactive minerals introduced by the manures were very limited, ruling out the possibility that

fertilizers introduced reactive fractions. Furthermore, Huang et al. (2016) indicated that organic

fertilization increased the iron freeness index (i.e., $Fe_d/Fe_t$ ratio) when compared to no fertilization and

chemical fertilization by citrate-bicarbonate-dithionite (CBD) extraction, suggesting a high degree of

soil weathering in organic fertilization. Therefore, we suggest that organic fertilization treatments *in situ*

enhance reactive minerals by the transformation of minerals. This was supported by a simulated study

that the addition of oxalic acid to soil colloids could promote the transformation from Fe(III) to

ferrihydrite (Huang et al. 2016). Another previous report also indicated that the low-molecular-weight

(LMW) organic acid might incorporate into the network structure of SRO minerals, inhibiting further

growth of SRO minerals (Xu et al., 2010).

In addition, previous studies showed that the formation of highly reactive Al and Fe minerals could

greatly benefit the binding and potential preservation of SOC (Torn et al., 1997; Wen et al., 2014a; Xiao

et al., 2015). Especially, reactive Fe minerals might be responsible for the retention of aromatic C and

O-alkyl C in soils (Huang et al., 2016). Under favorable conditions, SOC turnover in soils with highly

reactive Al and Fe minerals could persist in tephra beds for at least 250,000 yrs (Parfitt, 2009). The

accumulation of highly reactive Al and Fe minerals in soils could therefore improve SOC sequestration

under long-term organic manure fertilization. Furthermore, soil colloids usually consist of mixtures or

complexes of hydrous oxides of Fe, Al and natural organic matter, which have important implications

for deposition, aggregation, and sorption processes (Schumacher et al., 2005; Herrmann et al., 2007;

Mueller et al., 2012).

### 4.3. Environmental implications and technical challenges

Soils are highly complex materials that are structurally and elementally heterogeneous across a

wide range of spatial and temporal scales (Herrmann et al., 2007; Mueller et al., 2012; Vogel et al.,

2014). In porous media the stability, transport, and deposition of colloids, which usually consist of mixtures or complexes of hydrous oxides of Fe, Al, and natural organic matter, are strongly affected by the mobilized colloidal particles and specific surface area (Kaiser and Guggenberger, 2003; Schumacher et al., 2005). By combining HRTEM, NanoSIMS, XPS and/or XANES techniques, the present study investigated the previously unknown highly reactive mineral elements and their spatial distribution patterns under contrasting fertilizations. This strategy has the following key advantages: HRTEM, NanoSIMS images and elemental mapping with sufficient resolution are able to illustrate the specific relationship and spatial heterogeneity of organic, mineral complexes under contrasting fertilizations, while the decomposition of XPS and Fe K-edge XANES peaks to definite semi-quantitative determinations shows the elemental valence states and compositions. Nevertheless, we are still faced with the challenge of how to utilize spatial information to parameterize models for handling the complex, stochastic interactions between organo-mineral complexes and their microenvironments, including a range of biogeochemical transformation influenced by different fertilization treatments at the submicron scale (Remusat et al., 2012; Abdala et al., 2015; Hatton et al., 2015). Because of the highly heterogeneous distribution of mineral elements/C-functional groups in soils, investigations on more regions in more samples is necessary to obtain solid relationships between organic C and mineral elements using the NanoSIMS *in situ* observation. Meanwhile, an inadequate sample preparation to avoid artefacts is also a challenge, which may introduce a bias in the interpretation of NanoSIMS data and location of regions-of-interest (Herrmann et al., 2007). In addition, the complexity of iron chemistry in soils also makes the Fe XANES and EXAFS characterization a challenge. For example, the accuracy

of the LCF results is strongly affected by the correctness of the applied set of predictor variables (Prietzel et al., 2007). And EXAFS only provides average structural information over a short-range order, therefore it fails to determine if the minerals are crystalline or amorphous (Li et al., 2015), which is important in understanding the stabilization of organic carbon. With the enough soil samples and the improvement of sample preparation, these limitations can be well overcome, and the combination of

HRTEM, NanoSIMS, XPS and/or XANES technique is expected to receive wide applications in the fields of agricultural science, environmental science, and ecology science.

## 5. Conclusions

In this study, we showed that 24-year long-term (1990-2014) organic fertilization increased the carbon binding loading and the potential preservation capacity of soil colloids at the submicron scale.

These submicron scale findings suggest that both reactive mineral species and their associations with C are differentially affected by inorganic and organic fertilization. This may be attributed to a greater concentration of highly reactive Al and Fe minerals presenting under NPKM than under NPK. Meanwhile, we also demonstrate that the combination of nano-scale secondary ion mass spectrometry (NanoSIMS), high resolution-transmission electron microscopy (HRTEM), X-ray absorption fine

structure spectroscopy (XAFS), and X-ray photoelectron spectroscopy (XPS), is a promising strategy to distinguish relationships between C preservation and minerals in natural soil colloids as well as the potential for SOM accumulation under inorganic and organic fertilizations at the submicron scale. The strategy paves the way toward in situ characterization of organo-mineral associations, which is critical in understanding their associated SOM accumulation and soil carbon storage.

**Supplementary material related to this article is available online at: http://www.biogeosciences.net/**

**Acknowledgment.** The authors thank B.R. Wang for his assistance in soil sampling in the Qiyang Long-term Fertilization Station. We also thank the staffs from the 1W1B beamline at Beijing Synchrotron Radiation Facility, for assistances in data collection. This work was jointly financially

supported by National Natural Science Foundation of China (41371248 and 41371299), Natural Science Foundation of Jiangsu Province of China (BK20131321), the Qing Lan Project, the Innovative Research Team Development Plan of the Ministry of China (IRT1256), the 111 Project (B12009), the Priority Academic Program Development (PAPD) of Jiangsu Higher Education Institutions, and Research Project of Shanghai Municipal Bureau of Quality and Technical Supervision (2014-02).

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

**Figure Captions**

**Fig. 1.** High-resolution transmission electron microscopy (HRTEM) images of highly reactive minerals in colloids extracted from soil (Ferralic Cambisol) after 24-year long-term (1990-2014) NPKM fertilization. (a), TEM images; (b), HRTEM images and selected area electron diffraction (SAED) patterns of the two regions indicated by the blue squares, showing that the black region is a complete crystalline, while the grey region is amorphous; (c-d) energy dispersive X-ray analysis (EDX) images of the region 1 and region 2. NPKM, chemical NPK plus swine manure fertilization. Also see Fig. S2 for similar HRTEM images under the Control and NPK fertilizations.

**Fig. 2.** Representative NanoSIMS images of $^{12}C^-$, $^{27}Al^{16}O^-$ and $^{56}Fe^{16}O^-$ in soil colloids from three contrasting long-term (1990-2014) fertilization treatments (Control, no fertilization, $28 \times 28$ μm$^2$; NPK, chemical nitrogen, phosphorus and potassium fertilization, $30 \times 30$ μm$^2$; NPKM, chemical NPK plus swine manure fertilization, $25 \times 25$ μm$^2$). Note that the color intensity calibration bar displayed in the chemical maps corresponds to the relative concentrations of individual elements, but cannot be used to compare one element with another. Bar = 5 μm.

**Fig. 3.** Box plots of $^{12}C^-/^{27}Al^{16}O^-$ (a, b) and $^{12}C^-/^{56}Fe^{16}O^-$ (c, d) ratios reflecting the $^{12}C^-$ rich ROIs (a, c) and $^{12}C^-$ less rich ROIs (b, d) of the soil colloids from three contrasting long-term (1990-2014) fertilization treatments using NanoSIMS (for all spots). Control, no fertilization; NPK, chemical nitrogen, phosphorus and potassium fertilization; NPKM, chemical NPK plus swine manure fertilization. The $^{12}C^-$ rich ROIs include the areas above 90 pixels and the $^{12}C^-$ less rich ROIs include the areas in the range of 90-40 pixels under Control and NPK, which were above 50 pixels and in the range of 50-30

pixels under NPKM. The number n in figures represents the number of the selected ROIs. The line in the middle of the box is the median value and the square in the box is the mean value. The lines that protrude out of the boxes represent the 25th and 75th population percentiles. Outliers are shown as diamonds.

**Fig. 4.** XPS peak-fitting (Al $2p_{3/2}$ and C 1s) images recorded from soil (Ferralic Cambisol) colloids extracted under three long-term (1990-2014) fertilization treatments. Control, no fertilization; NPK, chemical nitrogen, phosphorus and potassium fertilization; NPKM, chemical NPK plus swine manure fertilization.

**Fig. 5.** Fe K-edge EXAFS (left) and Fourier transforms (right) of reference materials and soil colloids from three contrasting long-term (1990-2014) fertilization treatments. Control, no fertilization; NPK, chemical nitrogen, phosphorus and potassium fertilization; NPKM, chemical NPK plus swine manure fertilization.

**Table Captions**

**Table 1.** Basic physiochemical characteristics of soil samples from three contrasting long-term (1990-2014) fertilization treatments [a].

**Table 2.** Binding energy and quantitation/assignment of XPS spectral bands of soil samples from three contrasting long-term (1990-2014) fertilization treatments [a].

**Table 3.** Linear combination fit (LCF) results of Fe K-edge XANES spectra of the soil colloids from three contrasting long-term (1990-2014) fertilization treatments [a].

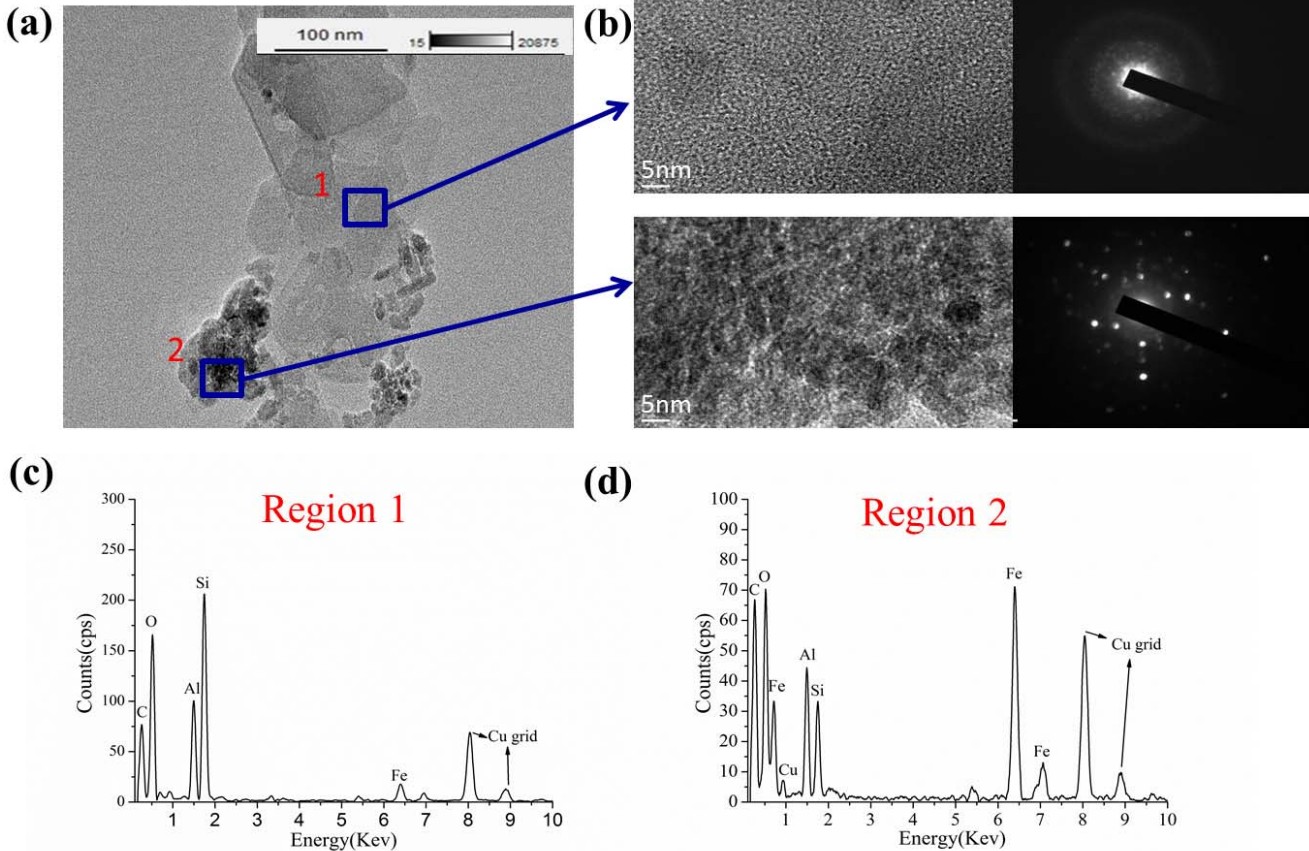

**Figure 1**

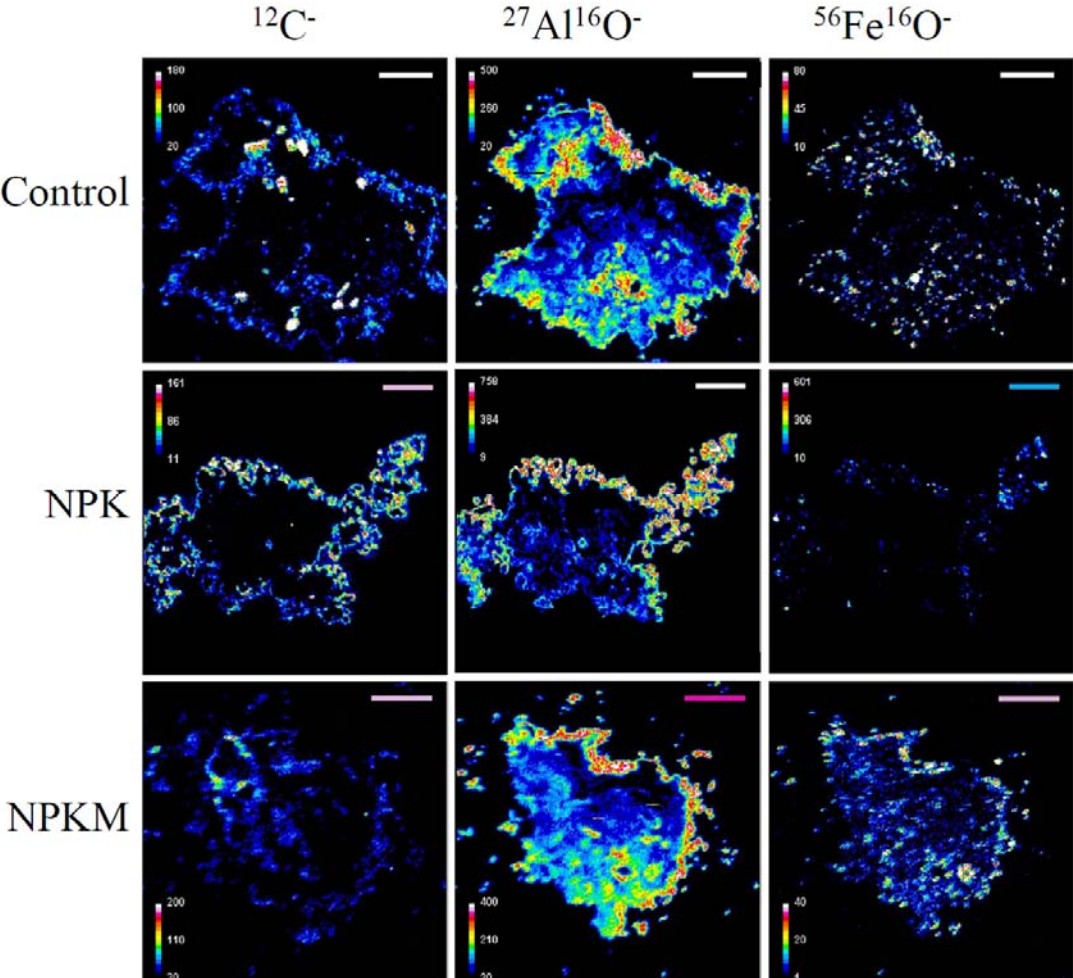

**Figure 2**

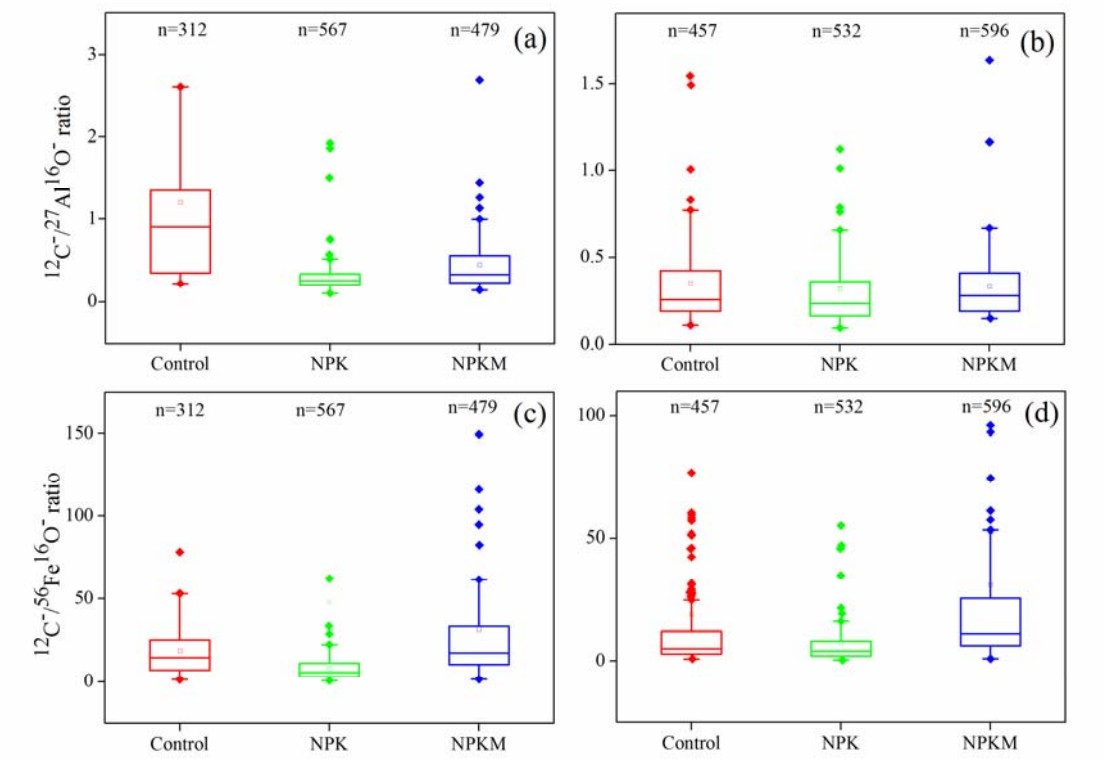

**Figure 3**

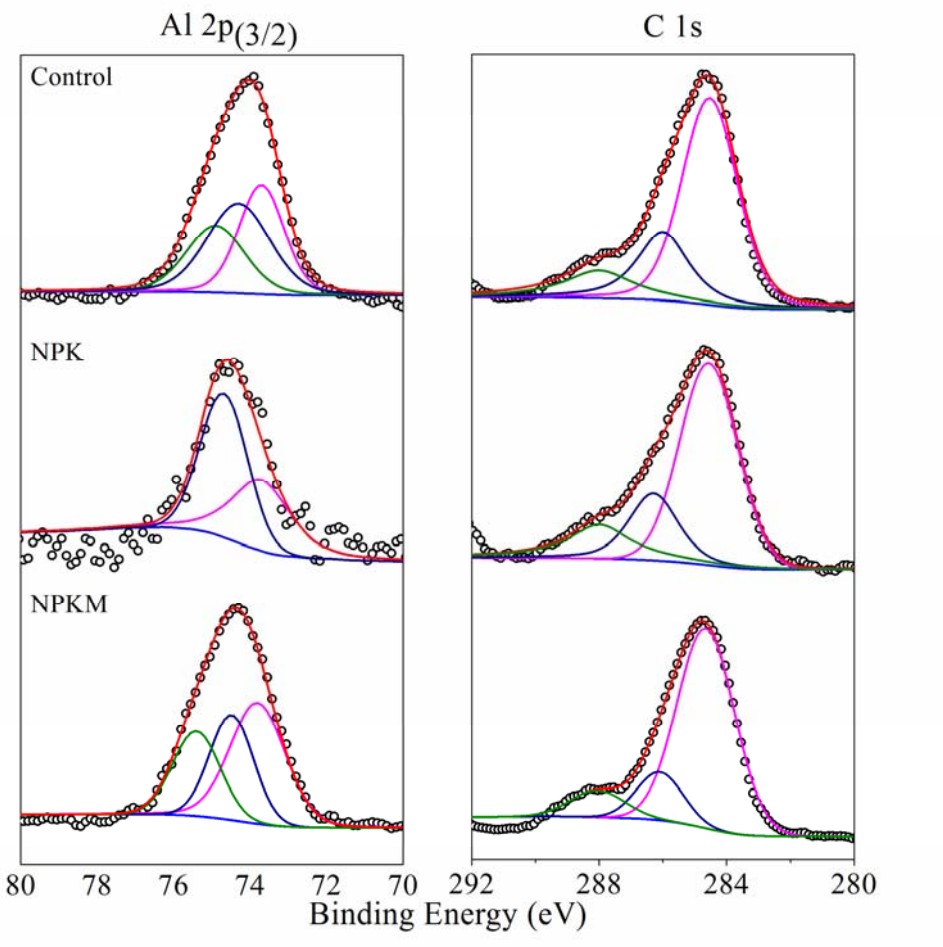

**Figure 4**

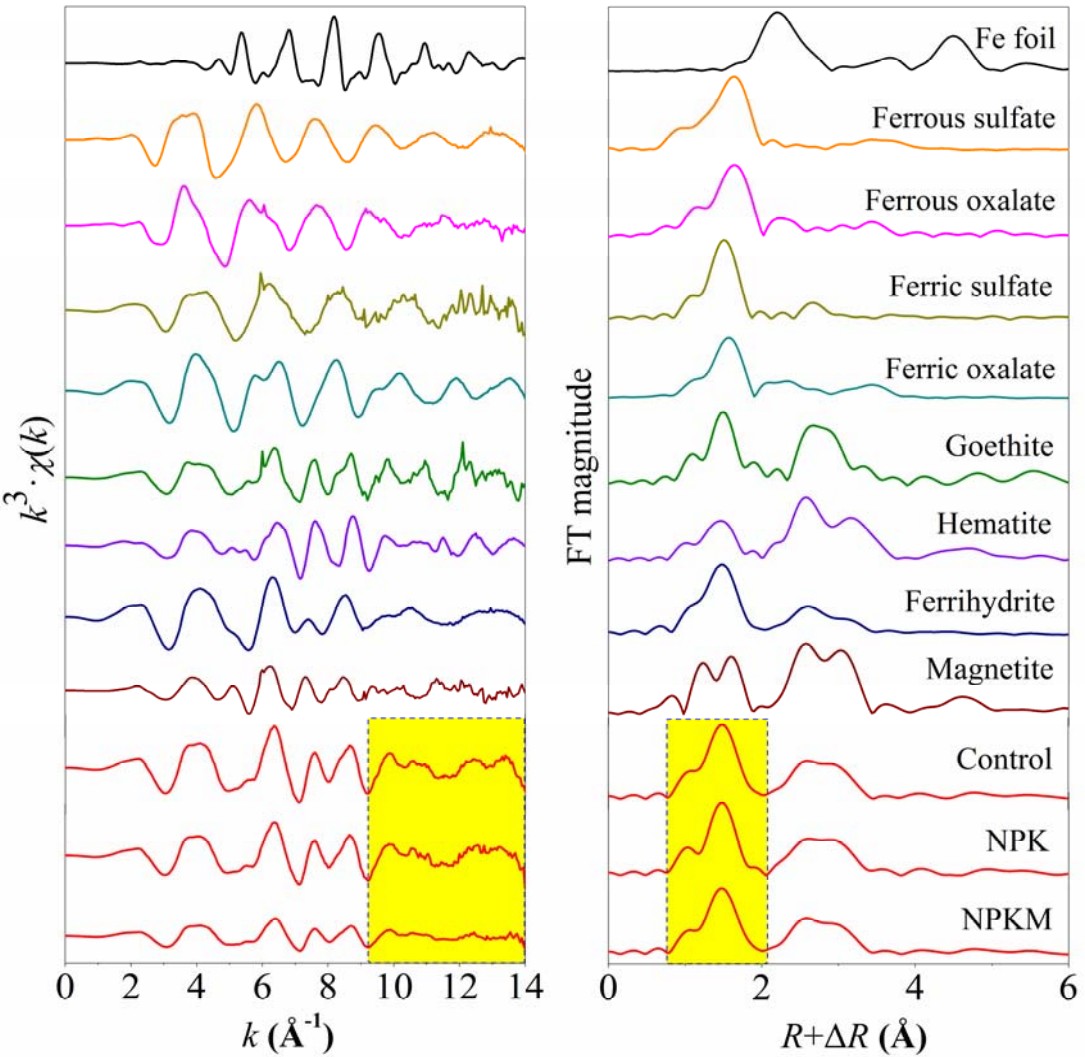

Fe foil

Ferrous sulfate

Ferrous oxalate

Ferric sulfate

Ferric oxalate

Goethite

Hematite

Ferrihydrite

Magnetite

Control

NPK

NPKM

$k^3 \cdot \chi(k)$

$k \ (\text{Å}^{-1})$

FT magnitude

$R + \Delta R \ (\text{Å})$

**Figure 5**

**Table 1. Basic physiochemical characteristics of soil samples from three separate long-term (1990-2014) fertilization treatments [a].**

| Treatment | Soil | | | | | Soil colloids | | | | |
|---|---|---|---|---|---|---|---|---|---|---|
| | Bulk soil pH (H$_2$O) | SOM (g kg$^{-1}$) | Al$_o$ (%) | Fe$_o$ (%) | SRO (%) | Al$_{XPS}$ (%) | Fe$_{XPS}$ (%) | DOC (mg L$^{-1}$) | DOC/Al$_{XPS}$ | DOC/Fe$_{XPS}$ |
| Control | 5.47 ± 0.04b | 14.88 ± 2.02c | 0.07 ± 0.003b | 0.20 ± 0.004b | 0.17 ± 0.00b | 6.23 | 1.47 | 6.17 | 0.99 | 4.20 |
| NPK | 4.15 ± 0.00c | 18.36 ± 0.16b | 0.04 ± 0.003c | 0.16 ± 0.003c | 0.12 ± 0.00c | 1.22 | 0.48 | 4.62 | 3.79 | 9.63 |
| NPKM | 5.84 ± 0.01a | 25.13 ± 2.02a | 0.11 ± 0.002a | 0.30 ± 0.007a | 0.26 ± 0.01a | 6.84 | 1.59 | 42.02 | 6.14 | 26.43 |

[a] Note: Control, no fertilization; NPK, chemical nitrogen, phosphorus and potassium fertilization; NPKM, chemical NPK plus swine manure fertilization, SOM, soil organic matter. Al$_{XPS}$ and Fe$_{XPS}$ indicated the surface concentration of Al and Fe in soil colloids, which were determined by the X-ray photoelectron spectroscopy (XPS). Al$_o$ and Fe$_o$ indicated reactive Al and Fe nanominerals, which were extracted using acid ammonium oxalate. DOC, dissolved organic carbon in soil colloids. Short-range ordered (SRO) minerals were calculated using the formula of Al$_o$ + 1/2 Fe$_o$ (%) (Kramer et al., 2012). Significant differences among fertilization treatments were determined using one-way ANOVA followed by the Tukey's HSD post hoc test at $P < 0.05$ after the conditions of normality and homogeneity of variance were met.

**Table 2. Binding energy and quantitation/assignment of XPS spectral bands of soil samples from three separate long-term (1990-2014) fertilization treatments [a].**

| Element | Control | | | NPK | | | NPKM | | |
|---|---|---|---|---|---|---|---|---|---|
| | Peak (eV) | Atomic (%) | Assignment | Peak (eV) | Atomic (%) | Assignment | Peak (eV) | Atomic (%) | Assignment |
| Al 2p$_{3/2}$ | 73.8 | 34.2 | Allophane $Al_2O_3$/ $Al_2O_3$–$nH_2O$ | 73.8 | 42.9 | Allophane $Al_2O_3$/ $Al_2O_3$–$nH_2O$ | 73.8 | 45.1 | Allophane $Al_2O_3$/ $Al_2O_3$–$nH_2O$ |
| Al 2p$_{3/2}$ | 74.3 | 39.0 | Boehmite AlO(OH) | 74.7 | 57.1 | Boehmite AlO(OH) | 74.5 | 29.4 | Boehmite AlO(OH) |
| Al 2p$_{3/2}$ | 74.9 | 26.8 | AlOx | / | / | / | 75.4 | 25.5 | AlOx |
| C 1s | 284.6 | 62.3 | Aromatic carbon (Ar-C-C/Ar-C-H) | 284.6 | 62.5 | aromatic carbon (Ar-C-C/Ar-C-H) | 284.6 | 75.9 | aromatic carbon (Ar-C-C/Ar-C-H) |
| C 1s | 286.1 | 23.6 | Ether or alcohol carbon (C-O) | 286.2 | 20.8 | Ether or alcohol carbon (C-O) | 286.1 | 14.7 | Ether or alcohol carbon (C-O) |
| C 1s | 288.0 | 14.2 | Ketonic or aldehyde carbon (C=O) | 288.0 | 16.7 | Ketonic or aldehyde carbon (C=O) | 288.0 | 9.5 | Ketonic or aldehyde carbon (C=O) |

[a] Note: Control, no fertilization; NPK, chemical nitrogen, phosphorus and potassium fertilization; NPKM, chemical NPK plus swine manure fertilization. The atomic percentage (%) is the corrected value calculated from the XPS peak-fitting areas (Childs et al., 1997; Crist, 2000) and elemental assignments were determined from published studies (Liang et al., 2008; Mikutta et al., 2009; Xiao et al., 2015).

**Table 3. Linear combination fit (LCF) results of Fe K-edge XANES spectra of the soil colloids from three separate long-term (1990-2014) fertilization treatments [a].**

| Treatment | LCF results (%) | | | | | | LCF parameters | |
|---|---|---|---|---|---|---|---|---|
| | Goethite | Hematite | Ferrihydrite | Ferric sulfates | Ferrous citrates | Ferrous sulfates | R-factor | Chi-square |
| **Control** | $66.0 \pm 0.025$ | $14.9 \pm 0.000$ | $16.0 \pm 0.025$ | ND | $3.10 \pm 0.012$ | ND | 0.000052 | 0.00437 |
| **NPK** | $67.0 \pm 0.025$ | $25.0 \pm 0.000$ | $6.30 \pm 0.020$ | ND | ND | $1.70 \pm 0.008$ | 0.000051 | 0.00426 |
| **NPKM** | $56.8 \pm 0.025$ | $20.4 \pm 0.000$ | $18.0 \pm 0.017$ | $4.8 \pm 0.018$ | ND | ND | 0.000051 | 0.00436 |

[a] Note: Control, no fertilization; NPK, chemical nitrogen, phosphorus and potassium fertilization; NPKM, chemical NPK plus swine manure fertilization. ND, not detected. Determination of parameters of fit (i.e., R-factor and chi-square) indicated that the LCF results are convincing.