# Peer review of "New strategies for submicron characterization the carbon binding of reactive minerals in long-term contrasting fertilized soils: Implications for soil carbon storage"

_Biogeosciences, 2015_

## Referee Comment (RC1) · Anonymous Referee #1 · 2 Mar 2016

The very interesting objective of the manuscript (ms) is that NKP + manure fertilization enhances the formation of reactive Fe and Al nano-minerals that may contribute to the soil carbon stabilization. The soil samples were collected from a 24 years fertilization field experiment and were compared to a control soil and a NPK fertilized soil. The authors applied a set of modern investigation techniques like NanoSIMS, HRTEM, SEAD, XANES, EXAFS and XPS to characterize the fractionated soil colloids. It remains however unclear in the ms if all these techniques used and the measured data are necessary for the achieved results and their rather poor discussion and the conclusion. The authors referred to some more and less relevant references in the discussion instead of a comprehensive interpretation of the results. In fact, this is a rather methodological paper and the (new) mechanistic insights are lacking.

Overall, some fundamental information about the manure application and SOC stocks could be included in the ms and useful for the discussion. (See also my comments below) The authors can find this information e.g. in a recent review paper by Maillard and Angers, 2014.

Manure addition enhanced the org C binding of Al and Fe nano-minerals which was also shown in the previous papers of the authors like in Wen et al. 2014 a and b. The authors should clearly present what was achieved in that papers and what is really new in the ms besides the combination of applied modern investigation techniques.

Title: What are interactive characteristics? What about the other methods used in the manuscript?

Introduction: The authors should add a paragraph about the effect of fertilization practices on SOM development of soils, especially about the effect of manure!

Line 115: ferrihydrite and allophanes.

2.6 EXAFS and XANES: details of the data processing are missing. How was the data fitting procedure? How were they normalized and in which interval? How many single scans were used?... criteria for the best fit?

Results: Line 207: significantly 217-219: the findings of SAED measurements are similar to the finding of Wen et al, 2014 paper. Why were the SAED experiments needed? What are the new knowledge and the benefit of these results for the ms?

Line 224: Fig. 1 and S2: not really clear what the authors want to demonstrate with the HRTEM and NMR (!!) spectra Line 230: What is the relevance of Fig. S2 and Table S2 for 12C- rich or less rich ROIs? Table S2 is about metal composition! See also line

232 Line 233: why is the area of percentage similar for the Control and the NPKM?

Lines 238-239: These results are expected. See paper Maillard and Angers, 2014. Line 241: present

Lines 245-251: the XANES results about Fe oxide species are interesting! But see my comments about EXAFS.

Line 252-253: see my comments for lines 238-239, these results are expected!

EXAFS results (lines 254-279): The authors confirm the findings of XANES. Why were both EXAFS and/or XANES measurements needed? What is the benefit of the EXAFS or/and XANES results?

XPS: (lines 280-285): interesting results! Can a direct uptake of manure be distinguished?

4.1. The discussion about the increased concentration of reactive Al and Fe minerals is missing in the chapter (the authors only cite previous works about this). Also a discussion of the org C content and composition is missing in lines 301-306.

4.2 : a discussion about the possible mechanism of the formation of reactive Al and Fe mineral is missing. It is further unclear why both XANES and EXAFS results are necessary and useful for the discussion.

The author included in the Discussion part some recent papers and their statements and repeated the advantages of the used methods but this did not result in a comprehensive discussion about the formation of reactive Fe and Al minerals and their role for the SOC stabilization.

---

## Author Comment (AC1) · 15 Mar 2016

Response to Referee 1

We thank Referee 1 for the thoughtful and supportive comments. We have revised our manuscript in response to the suggestions. All of the revised parts were colored in red in the revised manuscript.  Interactive comment on "In situ interactive characteristics of reactive minerals in soil colloids and soil carbon preservation differentially revealed

by nanoscale secondary ion mass spectrometry and X-ray absorption fine structure spectroscopy" by Jian Xiao et al.

Anonymous Referee #1

The very interesting objective of the manuscript (ms) is that NPK + manure fertilization enhances the formation of reactive Fe and Al nano-minerals that may contribute to the soil carbon stabilization. The soil samples were collected from a 24 years fertilization field experiment and were compared to a control soil and a NPK fertilized soil. The authors applied a set of modern investigation techniques like NanoSIMS, HRTEM, SEAD, XANES, EXAFS and XPS to characterize the fractionated soil colloids. It remains however unclear in the ms if all these techniques used and the measured data are necessary for the achieved results and their rather poor discussion and the conclusion. The authors referred to some more and less relevant references in the discussion instead of a comprehensive interpretation of the results. In fact, this is a rather methodological paper and the (new) mechanistic insights are lacking.

Response: In the revised manuscript, we add an explanation about these techniques, strengthening their complementary properties. Also, we strengthened the discussion by a comprehensive interpretation of the results. In addition, the conclusion section was also revised.

Overall, some fundamental information about the manure application and SOC stocks could be included in the ms and useful for the discussion. (See also my comments below) The authors can find this information e.g. in a recent review paper by Maillard and Angers, 2014.

Response: Good suggestion! In the revised manuscript, we added some fundamental information about the manure application and SOC stocks in the Introduction, including two papers about meta-analysis (Maillard and Angers, 2014; Zhao et al., 2015).

Manure addition enhanced the org C binding of Al and Fe nano-minerals which was

also shown in the previous papers of the authors like in Wen et al. 2014 a and b. The authors should clearly present what was achieved in that papers and what is really new in the ms besides the combination of applied modern investigation techniques.

Response: The results from our previous papers showed that manure amendments enhanced reactive components of minerals in soils, i.e., ferrihydrite and allophane. Because soil particles exhibit high heterogeneity, the in-situ characterization of the org C binding capability of Al and Fe nano-minerals at the submicron scale is urgent. To date, this information has not been explored in the contrasting fertilized soils. In the revised manuscript, we added this information in the Introduction section to strengthen the uniqueness of this manuscript.

Title: What are interactive characteristics? What about the other methods used in the manuscript? Response: In the revised manuscript, we removed "interactive" and the methods in the title. The title was changed to "New strategies for submicron characterization the carbon binding of reactive minerals in long-term contrasting fertilized soils: Implications for soil carbon storage".

Introduction: The authors should add a paragraph about the effect of fertilization practices on SOM development of soils, especially about the effect of manure!

Response: Based on the suggestion of Referee 1, we added a paragraph to describe the effect of fertilization practices (e.g., manure amendments) on SOM development of soils in the Introduction section.

Line 115: ferrihydrite and allophanes.

Response: In the revised manuscript, we changed "allophane and ferrihydrite" to "ferrihydrite and allophone" (Lines 117-118 in the revised manuscript).

2.6 EXAFS and XANES: details of the data processing are missing. How was the data fitting procedure? How were they normalized and in which interval? How many single scans were used?... criteria for the best fit?

Response: In the revised manuscript, we added these missing details, which can be seen at Lines 174-199 in the revised manuscript.

Results: Line 207: significantly 217-219: the findings of SAED measurements are similar to the finding of Wen et al, 2014 paper. Why were the SAED experiments needed? What are the new knowledge and the benefit of these results for the ms?

Response: In the revised manuscript, we changed "Significant higher SOM concentrations patterned as under NPKM > NPK than > Control (Table 1). Significantly higher percentages of oxalate extracted Al (Alo), Fe (Feo), SRO Al (Alxps ), Fe(Fexps ), DOC ranked as under NPKM > Control > NPK, but, significantly higher ratios of DOC/Alxps and DOC/Fexps were under NPKM > NPK > Control (Table 1)" (lines 207-211 in the original manuscript) to "SOM concentrations in different fertilization treatments ranked as NPKM >NPK> Control (Table 1). Oxalate extracted Al (Alo), Fe (Feo), SRO Al (Alxps), Fe (Fexps), and DOC ranked as NPKM > Control > NPK, but ratios of DOC/Alxps and DOC/Fexps ranked as NPKM > NPK > Control (Table 1)"(lines 215-218 in the revised manuscript).

The samples of SAED measurements in Wen et al (2014) paper were dissolved organic matters, which are different from the samples in this manuscript. In this manuscript, we used SAED to characterize soil colloids. By using SAED, we want to illustrate the element composition of mineral particles. Our results from SAED demonstrated that the amorphous mineral species were dominated by Al, Si, and O, while the crystalline minerals were mainly composed of Fe and O (Fig. 1-c, d). This information is meaningful for the following NanoSIMS analysis, supporting that it is suitable to focus on Al and Fe and their binding with org C.

Line 224: Fig. 1 and S2: not really clear what the authors want to demonstrate with the HRTEM and NMR (!!) spectra

Response: With the help of HRTEM (Fig. 1), we could directly visualize the differences in both morphology and elements compositions of soil particles, which is helpful for the

reason that we choose Al and Fe for binding org C in the following NanoSIMS. Fig. 2 shows the spatial heterogeneity of 12C-, 27Al16O-, and 56Fe16O- ion masses in the soil colloids. Fig. S2 presents the distribution of region of interests (ROIs), i.e., the 12C- rich ROIs and the 12C- less rich ROIs.

Line 230: What is the relevance of Fig.S2 and Table S2 for 12C- rich or less rich ROIs? Table S2 is about metal composition! See also line 232

Response: Fig.S2 was a representative NanoSIMS image showing the position of region of interests (ROIs) among the several replicates of different fertilization treatments as listed in Table S2 (Control: 8 replicates; NPK: 6 replicates; NPKM: 6 replicates, respectively). Table S2 presents the quantification of 12C- rich (12C–R) and 12C- less-rich (12C–LR) ROIs. In the revised manuscript, we added the more detail description about the Fig. S2 and Table S2.

Line 233: why is the area of percentage similar for the Control and the NPKM?

Response: The area of percentage for the Control and the NPKM is similar but different from the NPK, suggesting that compared to the Control treatment, the NPK treatment can change organo-mineral associations at the submicron scale in soil colloids, but NPKM can restore these associations.

Lines 238-239: These results are expected. See paper Maillard and Angers, 2014.

Response: We agree with the comments of Referee 1. These results in Lines 238-239 are expected, but they provided in-situ observation evidence at the submicron scale demonstrating that more organic C had been bound by Al and Fe minerals under NPKM than under NPK. This information has not been shown by previous publications. In the revised manuscript, we removed "the first" and rewrote the sentence.

Line 241: present

Response: In the revised manuscript, we changed "presented" to "present".

Lines245-251: the XANES results about Fe oxide species are interesting! But see my comments about EXAFS.

Response: The response about this question can be seen in the response of "EXAFS results".

Line 252-253: see my comments for lines 238-239, these results are expected!

Response: We agree with the comments of Referee 1. These results in Lines 252-253 are expected, but they provided in-situ observation evidence at the submicron scale. This information has not been shown by previous publications. In the revised manuscript, we added the comparison between this result and that from Maillard and Angers (2014).

EXAFS results (lines 254-279): The authors confirm the findings of XANES. Why were both EXAFS and/or XANES measurements needed? What is the benefit of the EXAFS or/and XANES results?

Response: XANES gives the quantitative composition of Fe minerals. Whereas EXAFS results can provide qualitative information about the structure of Fe minerals under different fertilization treatments. Furthermore, we use EXAFS results to show that the structure of Fe minerals under different fertilization treatments is really distinct. We believed that the EXAFS results are a helpful complementary to XANES. Combined XANES and EXAFS, we could get a complete understanding about the composition and coordination state of Fe minerals.

XPS: (lines 280-285): interesting results! Can a direct uptake of manure be distinguished?

Response: We really appreciate your interests. The previous investigation had shown that aromatic C in composted dairy manure accounted for approximately 30% of the total C, taking advantage of solid-state 13C nuclear magnetic resonance (NMR) spectroscopy (Liang et al., 1996). Another investigation showed that the addition of manurebased amendments, with or without chemical fertilizers, increased SOC and enhanced aggregate stability (Mikha et al., 2015). But it is unclear whether manure is direct contributed to aromatic C increase or first utilized by microbes and then contributed to aromatic C increase. In the revise manuscript, we added the above discussion in the Discussion section.

4.1. The discussion about the increased concentration of reactive Al and Fe minerals is missing in the chapter (the authors only cite previous works about this). Also a discussion of the org C content and composition is missing in lines 301-306.

Response: In the revised manuscript, we added the discussion about increased concentration of reactive Al and Fe minerals and the org C content and composition.

4.2: a discussion about the possible mechanism of the formation of reactive Al and Fe mineral is missing. It is further unclear why both XANES and EXAFS results are necessary and useful for the discussion. The author included in the Discussion part some recent papers and their statements and repeated the advantages of the used methods but this did not result in a comprehensive discussion about the formation of reactive Fe and Al minerals and their role for the SOC stabilization.

Response: In the revised manuscript, we added the discussion about the possible mechanism of the formation of reactive Al and Fe minerals. By measuring the composition of manure, Wen et al. (2014b) have shown that the reactive minerals introduced by the manures were very limited, ruling out the possibility that fertilizers introduced Al fractions. Therefore, we suggests that organic fertilization treatments enhance reactive minerals by the in-situ transformation of minerals. This is supported by the results from a simulated study that addition of oxalic acid to soil colloids can promote the transformation from Fe(III) to ferrihydrite (Huang et al., 2016). Another previous report also indicated that the low-molecular-weight (LMW) organic acid may incorporate into the network structure of SRO minerals, inhibiting further growth of SRO minerals (Xu et al., 2010).

In this study, XANES and EXAFS results provided quantitative and qualitative information about the composition of Fe minerals under different fertilization treatments, respectively. Both of them can support each other. Combined XANES and EXAFS, we could get a complete understanding about the composition and coordination state of Fe minerals. Moreover, we strengthened the discussion by a comprehensive interpretation of the results.

Please also note the supplement to this comment:
http://www.biogeosciences-discuss.net/bg-2015-625/bg-2015-625-AC1-supplement.zip

---

## Referee Comment (RC2) · Anonymous Referee #3 · 4 Apr 2016

I recommend the manuscript to be published after minor revisions. This is an excellent research that employ a combination of cutting-edge techniques (NanoSIMS, EXAFS) to explore the mineral-carbon association in soil colloids. The results would be of great significance to improve current understanding of the soil C pool and its stability towards global changes. However, I have two comments on the manuscript. First, the effect of long-term fertilization on the Al and Fe mineralogy was not fully discussed, although the nano-SIMS revealed that Fe(Al) and C are coupled. I wonder if chemical extraction

experiments could help address the changes of FeãĂĄC speciation during the long-term fertilization. Second, I would recommend the author put Fig. 5 to the supporting information, as they already have the EXAFS data in Fig. 6 and the Fe EXAFS data are believed to be more informative and quantitative than the XANES data.

---

## Author Comment (AC2) · 12 Apr 2016

Referee #3 Interactive comment on "In situ interactive characteristics of reactive minerals in soil colloids and soil carbon preservation differentially revealed by nanoscale secondary ion mass spectrometry and X-ray absorption fine structure spectroscopy" Jian Xiao et al. yuguanghui@njau.edu.cn

Response to Referee #3 We thank Referee #3 for the exciting and thoughtful com-

ments. We have revised our manuscript in response to the suggestions. All of the revised parts were colored in red in the revised manuscript. Interactive comment on "In situ interactive characteristics of reactive minerals in soil colloids and soil carbon preservation differentially revealed by nanoscale secondary ion mass spectrometry and X-ray absorption fine structure spectroscopy" by Jian Xiao et al.

Anonymous Referee #3 I recommend the manuscript to be published after minor revisions. This is an excellent research that employs a combination of cutting-edge techniques (NanoSIMS, EXAFS) to explore the mineral-carbon association in soil colloids. The results would be of great signifi-cance to improve current understanding of the soil C pool and its stability towards global changes. However, I have two comments on the manuscript. First, the effect of long-term fertilization on the Al and Fe mineralogy was not fully discussed, although the nano-SIMS revealed that Fe(Al) and C are coupled. I wonder if chemical extraction experiments could help address the changes of Fe, Al speciation during the long-term fertilization. Second, I would recommend the author put Fig. 5 to the supporting information, as they already have the EXAFS data in Fig. 6 and the Fe EXAFS data are believed to be more informative and quantitative than the XANES data.

Response: Great comments! Our previous studies have shown that long-term fertilization could affect the Al and Fe mineralogy using chemical extraction methods, including the acid ammonium oxalate extraction (Alo and Feo), pyrophosphate extraction(Alp and Fep) and citrate-bicarbon-ate-dithionite (CBD) solution (Fed) (Huang et al., 2016; Wen et al., 2014; Zhang et al., 2013). Specifically, Zhang et al. (2013) observed that the oxalate-extractable Fe (Feo) content of NPKM and M treatments was greater than that of N and NPK treatments in the 20–40 cm layer, but there was no statistical difference between the manure treatments (NPKM and M) and mineral fertilizer treatments (N and NPK) at 0–20 cm. Meanwhile, the pyrophosphate-extractable Fe (Fep) concentrations were less in the NPKM and M treatments than those in the N and NPK treatments at 0–20 cm. Using the citrate-bicarbonate-dithionite (CBD) extraction

method, Huang et al. (2016) showed that organic fertilization treatments (NPKM and M) increased the iron freeness index (i.e., the Fed/Fet ratio) when compared to chemical fertilization treatment (NPK). In addition, Wen et al. (2014) found that compared with chemical fertilization (N and NPK), organic fertilization (NPKM and M) significantly ($P < 0.05$) increased amorphous Al and decreased exchangeable Al, while the addition of lime (N with lime and NPK with lime) significantly ($P < 0.05$) increased weakly organically bound Al and decreased exchangeable Al. In the revised manuscript, we added the corresponding discussion and colored in red in the revised manuscript. Based on the suggestion of Referee 3, we had put Fig. 5 to the supporting information (as Fig. S3).

Please also note the supplement to this comment:
http://www.biogeosciences-discuss.net/bg-2015-625/bg-2015-625-AC2-supplement.pdf

———————————————————

**Control**

**(a)**

**(b)**

5nm

5nm

**(c)**

Region 1

**(d)**

Region 2

**Fig. 1.** High-resolution transmission electron microscopy (HRTEM) images of highly reactive minerals from colloids

[Figure]

**Fig. 2.** Representative NanoSIMS images of 12C-, 27Al16O- and 56Fe16O- in soil colloids from three contrasting long-term (1990-2014) fertilization treatments

[Figure]

**Fig. 3.** Box plots of 12C-/27Al16O- (a, b) and 12C-/56Fe16O- (c, d) ratios reflecting the 12C- rich ROIs (a, c) and 12C- 
[revised manuscript text omitted]

717 245-252, 2002.

---

## Referee Comment (RC3) · Anonymous Referee #4 · 4 May 2016

This paper, by Xiao et al., presents a study about the influence of fertilization (organic vs inorganic) on the colloidal interactions between soil organic matter and Al-/Fe- rich minerals. The authors used top end micro- and nano- scale techniques to characterize the mineralogy, the redox and the amount of organic matter in their sample.

The general topic of the study is well within the scope of biogeosciences.

I find this manuscript well written and that the study appears well designed. The abstract quality is good. The introduction provides a descent description of the scientific context of the study and presents the goal of this study. The sample and methods are well described. Nevertheless, I believe that the manuscript could really be improved before acceptation for publication. My main criticism is that I would expect the authors to discuss their data in more details. They seem to have acquired an impressive dataset, but the interpretation and discussion of the data, in addition to the insight we get from their comparison, is too short, to my sense. I'm sure there is much more to tell from their results. The authors should also present and discuss potential mechanistic processes that may explain their observations. Overall, this manuscript appears frustrating (we expect more in the discussion!).

In addition, I feel that some of the results should be presented in more details. For instance, the description of the NanoSIMS study, lines 208 to 211, is very short! I'm sure you have plenty of nice images. Please provide deeper description. Be more specific and indicate to the reader what the NanoSIMS brings to the study.

There are few additional points that should be clarified : - lines 114-116: "In this study, we chose 6 spots ..." unclear, should be rephrased. What do you mean when you write organo-mineral complexes were included? - line 128: you claim that depth resolution of the Cs beam is 15 nm. Where does it come from? Is it a calculation? Was it measured by anyone? Please add the source for this number. - line 131: it is ok to cite previous studies for details about analytical protocols, but the authors could at least provide their image size and resolution (i.e. number of pixels) as this is something that is adjusted from one study to the other. - lines 134 and following: the sorting in 12C rich and less rich areas is unclear, and the authors should explain why they have different conditions (limit at 90 or 50 pixels) depending on the sample. How does it give comparable results if the conditions to define areas are different? - line 142 and following: what do you mean by "the ROIs of the AIO and FeO images were combined..."? Please provide mode detail. Do you proceed this way to obtain a ROI corresponding to mineral rich regions? - line 270: I'm not really convinced that figure 3 shows what the authors claim.

---

## Author Comment (AC3) · 8 May 2016

Referee #4 Interactive comment on "In situ interactive characteristics of reactive minerals in soil colloids and soil carbon preservation differentially revealed by nanoscale secondary ion mass spectrometry and X-ray absorption fine structure spectroscopy" Jian Xiao et al. yuguanghui@njau.edu.cn

Response to Referee #4

We really thank Referee #4 for the exciting and thoughtful comments. We have revised our manuscript in response to these suggestions. All of the revised parts were colored in red in the revised manuscript. Interactive comment on "In situ interactive characteristics of reactive minerals in soil colloids and soil carbon preservation differentially revealed by nanoscale secondary ion mass spectrometry and X-ray absorption fine structure spectroscopy" by Jian Xiao et al.

Anonymous Referee #4

This paper, by Xiao et al., presents a study about the inïfluence of fertilization (organic vs inorganic) on the colloidal interactions between soil organic matter and Al-/Fe-rich minerals. The authors used top end micro- and nano- scale techniques to characterize the mineralogy, the redox and the amount of organic matter in their sample. The general topic of the study is well within the scope of biogeosciences. I find this manuscript well written and that the study appears well designed. The abstract quality is good. The introduction provides a descent description of the scientiïfic context of the study and presents the goal of this study. The sample and methods are well described. Nevertheless, I believe that the manuscript could really be improved before acceptation for publication. My main criticism is that I would expect the authors to discuss their data in more details. They seem to have acquired an impressive dataset, but the interpretation and discussion of the data, in addition to the insight we get from their comparison, is too short, to my sense. I'm sure there is much more to tell from their results. The authors should also present and discuss potential mechanistic processes that may explain their observations. Overall, this manuscript appears frustrating (we expect more in the discussion!).

Response: Great comments! In the revised manuscript, we strengthened the discussion. The revised parts were colored in red in the revised manuscript and not listed here for brevity.

In addition, I feel that some of the results should be presented in more details. For
instance, the description of the NanoSIMS study, lines 208 to 211, is very short! I'm sure you have plenty of nice images. Please provide deeper description. Be more specific and indicate to the reader what the NanoSIMS brings to the study.

Response: We agree with the comments of Referee 4! In the revised manuscript, we presented the results in more details. The revised parts were colored in red in the revised manuscript.

There are few additional points that should be clarified : - lines 114-116: "In this study, we chose 6 spots ..." unclear, should be rephrased. What do you mean when you write organo-mineral complexes were included?

Response: Thanks! In the revised manuscript, we rephrased this sentence. Also, we deleted "organo-mineral complexes were included". The revised parts were colored in red in the revised manuscript and also listed as follows.

"In this study, we chose 6 spots from the NanoSIMS images to show the replicates of each soil colloid sample because the majority of particulate organo-mineral complexes were included and similar according to the characterization of natural colloids (Philippe and Schaumann, 2014; Xiao et al., 2015)"

was changed to

"In this study, we achieved 6 NanoSIMS images for each soil colloid sample to support the replicates of our results (Philippe and Schaumann, 2014; Xiao et al., 2015)".

- line 128: you claim that depth resolution of the Cs beam is 15 nm. Where does it come from? Is it a calculation? Was it measured by anyone? Please add the source for this number.

Response: We appreciate the comment of Referee 4. Vogel et al. (2014) calculated that "The estimated depth resolution with 16 keV Cs+ ions as primary ion beam is about 15 nm." in their study. In the revised manuscript, we deleted this sentence after the discussion with one of the co-authors, Jialong Hao, who is responsible for the

measurement of NanoSIMS.

Vogel, C., Mueller, C. W., Hoschen, C., Buegger, F., Heister, K., Schulz, S., Schloter, M., and Kögel-Knabner, I.: Submicron structures provide preferential spots for carbon and nitrogen sequestration in soils, Nature communications, 5, 2947, 2014

- line 131: it is ok to cite previous studies for details about analytical protocols, but the authors could at least provide their image size and resolution (i.e. number of pixels) as this is something that is adjusted from one study to the other.

Response: We were sorry for the incomplete information. In the revised manuscript, we added the details. The revised parts were colored in red in the revised manuscript and also listed as follows.

"Specific details describing NanoSIMS measurements can be found in previous publications (Vogel et al., 2014; Xiao et al., 2015)."

was changed to

"In this study, the NanoSIMS images sizes were 256 × 256 pixels, Control, no fertilization, 28 × 28 $\mu$m2; NPK, chemical nitrogen, phosphorus and potassium fertilization, 30 × 30 $\mu$m2; NPKM, chemical NPK plus swine manure fertilization, 25 × 25 $\mu$m2, respectively. Other specific details describing NanoSIMS measurements can be found in previous publications (Vogel et al., 2014; Xiao et al., 2015)."

- lines 134 and following: the sorting in 12C rich and less rich areas is unclear, and the authors should explain why they have different conditions (limit at 90 or 50 pixels) depending on the sample. How does it give comparable results if the conditions to define areas are different?

Response: Good comments! This inspiration of sorting in 12C rich and less rich areas is from several excellent previous studies (Herrmann et al., 2007; Mueller et al., 2012; Rumpel et al., 2015; Vogel et al., 2014). For instance, Mueller et al. (2012) analyzed the spatial behavior of selected secondary ions along the line scans by choosing the

single ROIs comprised between 100 and 1500 pixels corresponding to 0.2-4 $\mu$m2 depending on particle size and ROI. Vogel et al. (2014) calculated the ROIs with an area greater than 10 pixels selected by the threshold option of the Image J software. In our study, the classification of sorting in 12C rich and less rich areas (limit at 90 or 50 pixels) were also based on the soil samples, which could represent the real intensity of ion mass. If we changed the conditions to a define area, the ratio between the 12C rich and less rich areas kept the same trend. In the revised manuscript, we added more details and marked them in red.

Reference

Herrmann, A. M., Ritz, K., Nunan, N., Clode, P. L., Pett-Ridge, J., Kilburn, M. R., Murphy, D. V., O'Donnell, A. G., and Stockdale, E. A.: Nano-scale secondary ion mass spectrometry-A new analytical tool in biogeochemistry and soil ecology: A review article, Soil Biology and Biochemistry, 39, 1835-1850, 2007.

Mueller, C. W., Kölbl, A., Hoeschen, C., Hillion, F., Heister, K., Herrmann, A. M., and Kögel-Knabner, I.: Submicron scale imaging of soil organic matter dynamics using NanoSIMS-From single particles to intact aggregates, Organic Geochemistry, 42, 1476-1488, 2012.

Rumpel, C., Baumann, K., Remusat, L., Dignac, M.-F., Barré, P., Deldicque, D., Glasser, G., Lieberwirth, I., and Chabbi, A.: Nanoscale evidence of contrasted processes for root-derived organic matter stabilization by mineral interactions depending on soil depth, Soil Biology and Biochemistry, 85, 82-88, 2015.

Vogel, C., Mueller, C. W., Hoschen, C., Buegger, F., Heister, K., Schulz, S., Schloter, M., and Kögel-Knabner, I.: Submicron structures provide preferential spots for carbon and nitrogen sequestration in soils, Nature communications, 5, 2947, 2014.

- line 142 and following: what do you mean by "the ROIs of the AlO and FeO images were combined..."? Please provide mode detail. Do you proceed this way to obtain a

ROI corresponding to mineral rich regions?

Response: We were sorry for the confusing writing. In this study, we sorted the ROIs into 12C rich and less rich areas according to the pixels of 12C ion mass. Meanwhile, the ROIs of AIO and FeO ion mass were similar as the 12C ion mass because we were interested in studying the in situ location and correlation between the organic and mineral compositions in soil samples.

In the revised manuscript, we changed "The ROIs of the 27Al16O- and 56Fe16O- images were combined afterwards to obtain the ROIs according to the distribution of the 12C- rich ROIs and 12C- less rich ROIs under different fertilizations conditions (Table S2)." to "The same ROIs were simultaneously selected in the 27Al16O- and 56Fe16O- images. "

- line 270: I'm not really convinced that ïñẮgure 3 shows what the authors claim.

Response: Sorry for the unclear statement. In the revised manuscript, we rephased this sentence. The revised parts were colored in red in the revised manuscript and also listed as follows.

"In this study, the ROI analyses of NanoSIMS in situ observation (Fig. 3) provided direct evidence that long-term organic fertilization strengthened the SOC binding and preservation capability of Al and Fe minerals in soil colloids as well as a highly spatial heterogeneity of soil colloids at the submicron scale."

was changed to

"In this study, the ROI analyses of NanoSIMS observation (Fig. 3) indicated that despite of highly spatial heterogeneity of organo-mineral complexes at the submicron scale, long-term organic fertilization strengthened the SOC binding and potential preservation capability of Fe minerals for both the 12C- rich- or 12C- less rich- ROIs in soil colloids when compared to chemical fertilization. Meanwhile, long-term organic fertilization also strengthened the SOC binding and potential preservation capability

of Al minerals for the 12C- rich- ROIs in soil colloids when compared to chemical fertilization. However, as for 12C- less rich- ROIs in soil colloids, fertilization regimes seemed to have no influence in the SOC binding with Al minerals."

Please also note the supplement to this comment:
http://www.biogeosciences-discuss.net/bg-2015-625/bg-2015-625-AC3-supplement.zip

---

## Author Response (AR1)

**yuguanghui@njau.edu.cn**

**Associate Editor Decision: Publish subject to minor revisions (Editor review) (26 May 2016) by Dr. Roland Bol**

Comments to the Author:

Figure 1 exist of 12 sub figures, in the text it is not clear what specific sub figures is referred too when all 3 treatments have a, b, c and d. One should use individual letter for each sub figure.

The authors may want to considering putting some of these 12 sub figures into supplementary material and only show the key sub figures.

The authors should in the revised version implement all what is promised in response to the posed comments in the interactive discussion and a final check of the English be useful.

**Response to Associate Editor**

Comment1: Figure 1 exist of 12 sub figures, in the text it is not clear what specific sub figures is referred too when all 3 treatments have a, b, c and d. One should use individual letter for each sub figure.

The authors may want to considering putting some of these 12 sub figures into supplementary material and only show the key sub figures.

**Response 1:** Many thanks for the above-mentioned suggestions from the Associate Editor. In the revised manuscript, we only presented the results of high-resolution transmission electron microscopy (HRTEM) images of colloids extracted from the NPKM fertilized soil. And the HRTEM images of the control- and NPK- fertilized soils have been put into supplementary materials as Fig. S2, and the two sets of "a, b, c and d" have been changed to "a, b, c, d for the sub-figures under Control and e, f, g and h for the sub-figures under NPK" in the new Fig. S2, accordingly.

Comment 2: The authors should in the revised version implement all what is promised in response to the posed comments in the interactive discussion and a final check of the English be useful.

**Response 2:** In the revised manuscript, we have implemented all valuable comments into the manuscript and carefully checked the English writings. The implemented comments and the revised areas have been presented in red color in the revised manuscript.

**Marked Manuscript Version**

[revised manuscript text omitted]
 | $5.47 \pm 0.04$b | $14.88 \pm 2.02$c | $0.07 \pm 0.003$b | $0.20 \pm 0.004$b | $0.17 \pm 0.00$b | 6.23 | 1.47 | 6.17 | 0.99 | 4.20 |
| NPK | $4.15 \pm 0.00$c | $18.36 \pm 0.16$b | $0.04 \pm 0.003$c | $0.16 \pm 0.003$c | $0.12 \pm 0.00$c | 1.22 | 0.48 | 4.62 | 3.79 | 9.63 |
| NPKM | $5.84 \pm 0.01$a | $25.13 \pm 2.02$a | $0.11 \pm 0.002$a | $0.30 \pm 0.007$a | $0.26 \pm 0.01$a | 6.84 | 1.59 | 42.02 | 6.14 | 26.43 |

[a] Note: Control, no fertilization; NPK, chemical nitrogen, phosphorus and potassium fertilization; NPKM, chemical NPK plus swine manure fertilization, SOM, soil organic matter. Al$_{XPS}$ and Fe$_{XPS}$ indicated the surface concentration of Al and Fe in soil colloids, which were determined by the X-ray photoelectron spectroscopy (XPS). Al$_o$ and Fe$_o$ indicated reactive Al and Fe nanominerals, which were extracted using acid ammonium oxalate. DOC, dissolved organic carbon in soil colloids. Short-range ordered (SRO) minerals were calculated using the formula of Al$_o$ + 1/2 Fe$_o$ (%) (Kramer et al., 2012). Significant differences among fertilization treatments were determined using one-way ANOVA followed by the Tukey's HSD post hoc test at $P < 0.05$ after the conditions of normality and homogeneity of variance were met.

665

**Table 2. Binding energy and quantitation/assignment of XPS spectral bands of soil samples from three separate long-term (1990-2014) fertilization treatments [a].**

| Element | Control | | | NPK | | | NPKM | | |
|---|---|---|---|---|---|---|---|---|---|
| | Peak (eV) | Atomic (%) | Assignment | Peak (eV) | Atomic (%) | Assignment | Peak (eV) | Atomic (%) | Assignment |
| Al 2p$_{3/2}$ | 73.8 | 34.2 | Allophane $Al_2O_3$/ $Al_2O_3$–$nH_2O$ | 73.8 | 42.9 | Allophane $Al_2O_3$/ $Al_2O_3$–$nH_2O$ | 73.8 | 45.1 | Allophane $Al_2O_3$/ $Al_2O_3$–$nH_2O$ |
| Al 2p$_{3/2}$ | 74.3 | 39.0 | Boehmite AlO(OH) | 74.7 | 57.1 | Boehmite AlO(OH) | 74.5 | 29.4 | Boehmite AlO(OH) |
| Al 2p$_{3/2}$ | 74.9 | 26.8 | AlOx | / | / | / | 75.4 | 25.5 | AlOx |
| C 1s | 284.6 | 62.3 | Aromatic carbon (Ar-C-C/Ar-C-H) | 284.6 | 62.5 | aromatic carbon (Ar-C-C/Ar-C-H) | 284.6 | 75.9 | aromatic carbon (Ar-C-C/Ar-C-H) |
| C 1s | 286.1 | 23.6 | Ether or alcohol carbon (C-O) | 286.2 | 20.8 | Ether or alcohol carbon (C-O) | 286.1 | 14.7 | Ether or alcohol carbon (C-O) |
| C 1s | 288.0 | 14.2 | Ketonic or aldehyde carbon (C=O) | 288.0 | 16.7 | Ketonic or aldehyde carbon (C=O) | 288.0 | 9.5 | Ketonic or aldehyde carbon (C=O) |

[a] Note: Control, no fertilization; NPK, chemical nitrogen, phosphorus and potassium fertilization; NPKM, chemical NPK plus swine manure fertilization. The atomic percentage (%) is the corrected value calculated from the XPS peak-fitting areas (Childs et al., 1997; Crist, 2000) and elemental assignments were determined from published studies (Liang et al., 2008; Mikutta et al., 2009; Xiao et al., 2015).

**675** **Table 3. Linear combination fit (LCF) results of Fe K-edge XANES spectra of the soil colloids from three separate long-term (1990-2014) fertilization treatments [a].**

| Treatment | LCF results (%) | | | | | | LCF parameters | |
|---|---|---|---|---|---|---|---|---|
| | Goethite | Hematite | Ferrihydrite | Ferric sulfates | Ferrous citrates | Ferrous sulfates | R-factor | Chi-square |
| **Control** | $66.0 \pm 0.025$ | $14.9 \pm 0.000$ | $16.0 \pm 0.025$ | ND | $3.10 \pm 0.012$ | ND | 0.000052 | 0.00437 |
| **NPK** | $67.0 \pm 0.025$ | $25.0 \pm 0.000$ | $6.30 \pm 0.020$ | ND | ND | $1.70 \pm 0.008$ | 0.000051 | 0.00426 |
| **NPKM** | $56.8 \pm 0.025$ | $20.4 \pm 0.000$ | $18.0 \pm 0.017$ | $4.8 \pm 0.018$ | ND | ND | 0.000051 | 0.00436 |

[a] Note: Control, no fertilization; NPK, chemical nitrogen, phosphorus and potassium fertilization; NPKM, chemical NPK plus swine manure fertilization. ND, not detected. Determination of parameters of fit (i.e., R-factor and chi-square) indicated that the LCF results are convincing.